# Spatial autocorrelation and determinants of low uptake of breast cancer screening among women of reproductive age: A mixed-effect multilevel analysis of Tanzanian population-based survey

**Deogratius Bintabara**[1]*, **Costantine C. Kamata**[2], **Ramadhani Mohamedi**[3], **Namanya Basinda**[4]

1 Department of Community Medicine, School of Medicine and Dentistry, The University of Dodoma, Dodoma, Tanzania, 2 Department of Anatomy and Histology, School of Medicine and Dentistry, The University of Dodoma, Dodoma, Tanzania, 3 Department of Information Systems and Technology, College of Informatics and Virtual Education, The University of Dodoma, Dodoma, Tanzania, 4 Department of Community Medicine, School of Public Health, Catholic University of Health and Allied Sciences, Mwanza, Tanzania

* bintabaradeo@gmail.com

## Abstract

### Introduction

Breast cancer remains an important public health problem with high mortality in low-income countries like Tanzania. This is because of the low uptake of screening for breast cancer, an intervention that could be cost-effective and significant in reducing mortality and poor prognosis in such a setting. This is a population-based survey to uncover the spatial distribution and determinants of low uptake of breast cancer screening among women of reproductive age in Tanzania.

### Methods

This analytical cross-sectional study utilized data from 2022 Tanzania Demographic and Health Survey and Malaria Indicator Survey (TDHS-MIS). A total of 15254 women aged 15–49 years were included in the analysis. The outcome variable was the uptake of breast cancer screening, coded as "1" for the women who reported a doctor or other healthcare provider examined their breasts to check for cancer, and "0" otherwise. Descriptive and geospatial analyses were conducted to assess patterns of screening uptake across regions. To identify associated factors, a mixed-effect multilevel logistic regression analysis was performed using Stata version 17. Adjusted odds ratios (AORs) with 95% confidence intervals (CIs) were reported, and a significance level of $p < 0.05$ was used.

**Data availability statement:** All relevant data are within the manuscript and its Supporting Information files.

**Funding:** The author(s) received no specific funding for this work.

**Competing interests:** The authors have declared that no competing interests exist.

## Results

The findings revealed that only 5% of the respondents reported having undergone breast cancer screening in Tanzania. The lowest uptake was in the Western (2.17%) and Southern (3.34%) zones of Tanzania. Regions with the poorest uptake of breast cancer screening were Kigoma (1.68%), Katavi (1.94%), Singida (1.54%), and Tabora (1.66%). Women with older ages, formal education, health insurance coverage, and reading newspapers, magazines, or using the internet had higher odds of uptake breast cancer screening than their counterparts.

## Conclusions

The uptake of breast cancer screening remains low throughout Tanzania and worst situation was noted in rural areas. Formal education, insurance coverage, and access to information continue to propel the low uptake of breast cancer screening in Tanzania. The fair distribution of health-promotive services could be vital in increasing uptake of breast cancer screening for the early detection and prevention of mortalities and other outcomes related to this severe disease.

## Introduction

Breast cancer remains a significant public health problem, being both the most common and a leading cause of cancer-related deaths among women globally [1]. In 2022 it accounted for 2.3 million new cancer cases, making it the second most diagnosed after lung cancer which had 2.5 million cases [2]. In terms of mortality, BC ranked fourth, with 670000 deaths worldwide, trailing behind lung, colorectal, and liver cancers which were responsible for 1.8 million, 900,000, and 760,000 deaths, respectively [1,2]. Despite lower incidence rates, breast cancer mortality rates are disproportionately higher in Low and Middle-Income Countries (LMICs) than in High-Income Countries (HICs) [3]. This is potentially due to late diagnosis and limited access to effective targeted therapies. In Tanzania, Breast cancer ranks second only to cervical cancer in both incidence and mortality among women. In 2018, it was estimated that 3,037 new cases and 1,303 deaths occurred, and projections indicate that both incidence and mortality could increase by more than 120% by 2040 [4–6]. This alarming trend underscores the urgent need for early detection and strategic public health interventions to mitigate the growing burden of breast cancer in the country.

Early detection, especially through regular screening is crucial in improving treatment outcomes and survival rates for most cancers [7–9]. However, in many LMICs, including Tanzania, the uptake of breast cancer screening services, is unsatisfactorily low despite its proven effectiveness in reducing morbidity and mortality [10,11]. Although governments and stakeholders have made efforts to expand access to both organized and opportunistic screening services, many patients still present with late-stage diseases [12,13]. Such cases are often difficult to manage effectively due to limited availability of specialized cancer care services and a shortage of advanced

diagnostic and therapeutic technologies [14–16]. In response to this growing burden, the Tanzanian government has introduced several policy measures to promote early detection and timely treatment. The Ministry of Health's *National Guidelines for Early Diagnosis of Breast Cancer and Referral for Treatment*, released in 2018 [17], provide standardized protocols for clinical breast examination, referral systems, and health worker training. Furthermore, collaborative initiatives involving the Ocean Road Cancer Institute and various non-governmental organizations have supported awareness campaigns, mobile outreach clinics, and educational programs targeting communities. Despite these efforts, implementation remains uneven, particularly in rural and hard-to-reach areas, underscoring the need for improved funding, infrastructure, and coordination to ensure equitable access to screening and diagnostic services across the country.

Several studies in sub-Saharan Africa have identified multiple factors associated with the low uptake of breast cancer screening. Key individual-level factors include limited awareness and knowledge about breast cancer and its screening methods, misconceptions about personal risk, and fear of a positive diagnosis, which often deters women from seeking early detection services [6,18,19]. Socio-cultural influences such as stigma surrounding cancer, reliance on traditional medicine, gender norms limiting women's autonomy in health decision-making, and community-level misconceptions have also been reported [18–20]. Religious beliefs may contribute to fatalistic attitudes, with some individuals perceiving cancer as a spiritual punishment or relying solely on faith-based healing [19,21]. Economically, the cost of transportation, screening fees, and loss of income due to clinic visits present significant obstacles, particularly for low-income and rural populations [22,23]. In addition, structural challenges such as a lack of nearby screening facilities and inadequately trained healthcare providers further hinder access and utilization of breast cancer screening services [20,24]. In Tanzania, several studies have assessed the uptake of breast cancer screening with many conducted at the health facility or regional level, thereby limiting their generalizability for national policy planning [20,22,24,25]. While a recent national-level analysis using the 2022 TDHS-MIS data have examined breast cancer screening in Tanzania and identified individual-level determinants [26,27]. While informative, these analyses did not evaluate spatial variation in screening nor quantify community-level effects using multilevel models. Therefore, gaps remain in understanding the spatial distribution and multilevel determinants of uptake of breast cancer screening.

To address this gap, the current study examines the spatial distribution and multilevel determinants of low uptake of breast cancer screening among women of reproductive age in Tanzania. By utilizing data from the 2022 TDHS-MIS, we employ a spatial analysis approach to identify geographic disparities in screening uptake. This spatial perspective is essential for highlighting regions with particularly low service utilization, thereby guiding resource allocation and targeted interventions. Furthermore, we apply a mixed-effect multilevel analysis approach to capture individual-, household-, and community-level factors influencing uptake of breast cancer screening. Recognizing that screening uptake is shaped by both personal characteristics and broader contextual determinants, this comprehensive approach provides deeper insights into how interventions can be tailored across different population strata and geographic settings.

By identifying both the spatial distribution and multilevel determinants of breast cancer screening uptake, the findings of this study aim to support public health efforts in developing context-specific and geographically targeted strategies to improve early detection and reduce breast cancer mortality in Tanzania and other countries with similar contexts.

## Methods

### Data source

This study analysed the data from the 2022 TDHS-MIS. This 2022 TDHSMIS was conducted by the National Bureau of Statistics (NBS) and the Office of the Chief Government Statistician Zanzibar (OCGS) in collaboration with the Ministries of Health (MoH) in Tanzania Mainland and Zanzibar. The Tanzania Food and Nutrition Centre (TFNC) collaborated on several aspects of the survey, especially biomarkers. Data collection took place from February to July 2022. The technical support for the survey was provided by ICF International under the DHS program. These surveys have been conducted after every four years since 1990. The survey collected information from a nationally representative sample of women aged 15–49 years in the selected households.

## Study design

This study employed analytical cross-sectional design based on data from a recent nationwide population-based survey. The analysis utilized information collected through interviews with women of reproductive age who were either residents or visitors in the selected households on the night preceding the survey.

## Sample size and sampling procedure

The 2022 TDHS-MIS employed stratified, two-stage cluster sampling design to obtain a nationally representative sample. The sampling was designed to produce reliable estimates at the national level, for urban and rural areas separately within Tanzania Mainland, and for Zanzibar. In the first stage, 629 enumeration areas (EAs) were selected from the sampling frame established during the 2012 Tanzania Population and Housing Census. These EAs, also referred to as clusters, were selected using a probability proportional to size (PPS) method within each sampling stratum, ensuring that larger EAs had a higher chance of selection. In the second stage, 26 households were systematically selected from a complete listing of all households in each of the selected clusters. This process yielded a target sample of 16,354 households. However, one EA was not reached due to security reasons, resulting in a final selection of 16,312 households. Of these, 15,907 were found to be occupied at the time of the survey, an interview were successfully completed in 15,705 households. From these interviewed households, a total of 15,699 women aged 15–49 were identified as eligible for individual interviews. Among them, 15,254 women, yielding a response rate of 97%. These women formed analytical sample used in the current study. The rigorous sampling approach ensures that the survey results are representative of the broader population across different geographic and socioeconomic strata in Tanzania (Fig 1).

## Data collection and processing

The TDHS-MIS used five questionnaires during data collection. However, this study used data collected using the Women's Questionnaire. After the pretesting of the questionnaires, the finalized and corrected version was used in the main surveys. Data collection was performed by trained and qualified interviewers through a series of practical tests and examinations. Data were collected by using computer-assisted personal interviewing (CAPI) from 24 February to 21 July 2022 (5 months). The data processing of the 2022 TDHS-MIS ran concurrently with the data collection exercise. The electronic data files from each completed cluster were transferred via Syncloud to the NBS central office server in Dodoma. The data files were registered and checked for inconsistencies, incompleteness, and outliers. Errors and inconsistencies were communicated to the field teams for review and correction. Finally, data editing and cleaning were performed.

## Measurement of variable

**Outcome variable.** The uptake of breast cancer screening was the outcome variable in this study. This variable was coded as "1" for 'Yes' if a woman reported being examined for a breast cancer by a doctor or other healthcare provider. The examination could include either a clinical breast examination, in which healthcare providers use their hands to feel for lumps or other changes, or the use of medical equipment to make an image of the breast tissue, such as a mammogram. Otherwise, coded as "0" for 'No' if she was not examined.

**Independent variables.** Following a detailed literature review on factors associated with the uptake of preventive services for women of reproductive age, a total of thirteen independent variables available in 2022 TDHS-MIS were included in this analysis. These variables were grouped into individual or household variables (level 1) and cluster variables (level 2). The level 1 variables were age (15–29, 30–34, 35–39, 40–49), marital status (never married, married/living together, separated/widow/divorced), education level (none, primary, secondary, tertiary), employed in last 12 months (not employed, employed for cash, employed but paid-in-kind), covered by health insurance (yes, no), parity (nullipara, primipara, multipara), reading newspaper/magazine (yes, no), listening to radio (yes, no), watching television

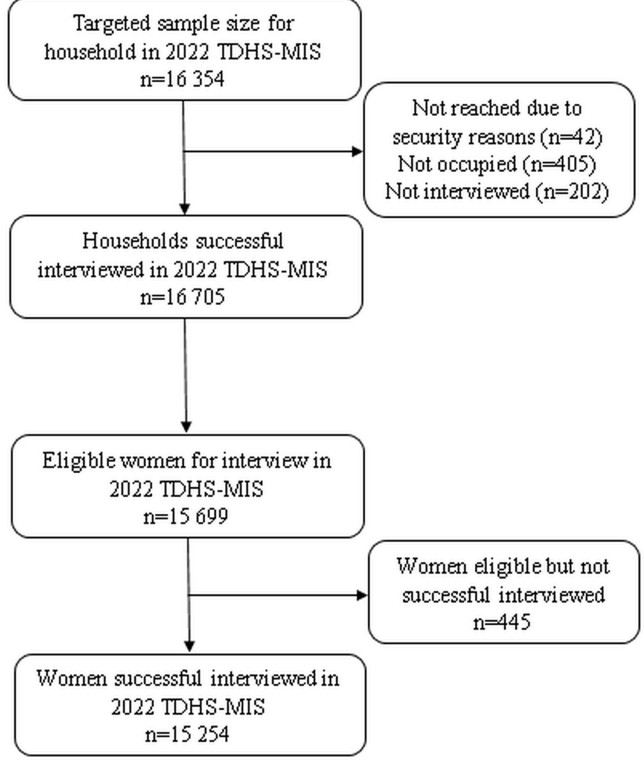

**Fig 1. Selection procedure for the sampling units included in this analysis.**

(yes, no), use of internet in last 12 months (yes, no) and household wealth status (poor, middle, rich). The level 2 variables were geographical zone (central, coastal, lake, northern, southern, southern highlands, western, Zanzibar) and residence (rural, urban). The selection of these variables and their categorization were based on previous studies [28–30].

## Statistical analysis

Descriptive analysis, geospatial analysis, and mixed-effect multilevel logistic regression analysis have been performed in this study. In descriptive analyses, all categorical variables were summarized using frequencies and percentages and then presented in either tables or graphs. Continuous variables, such as age and parity, were summarized using measures of the central tendency (median) and dispersion (interquartile range (IQR)) due to their non-normal distribution. Normality was assessed using the Shapiro-Wilk test, which indicated significant deviation from normality ($p < 0.05$), justifying the use of these non-parametric summary measures. The mixed-effects multilevel logistic regression analysis was performed to account for the hierarchical structure of the data, whereby individuals or households (level 1) were nested within clusters (level 2) refer to communities or neighborhoods within which households were surveyed. The multilevel models with two levels were fitted to identify factors associated with the uptake of breast cancer screening. A total of four models with the outcome variable "uptake of breast cancer screening" were estimated. In Model I (empty model) no independent variable was added. This model showed the changes in the uptake of breast cancer screening attributed to variations between clusters. In Model II, the individual-level variables were included, while in Model III, the cluster-level variables were included. In Model IV, both individual or household variables and cluster variables were included. The mixed-effect regression analysis provided results that included both fixed effects and random effects. The fixed effect results from

Model II, III, and IV were presented as odds ratios (OR) with their corresponding confidence interval (CI), while the results of the random effects were presented as the variance which indicates variations in uptake of breast cancer screening. The variance random component of the models was estimated by calculating the variance of the cluster-level variations and their corresponding standard errors. The Intra-class correlation (ICC) was calculated to evaluate whether the variations in the uptake of breast cancer are primarily within or between the clusters. Since the individual or household (level 1) was nested within the clusters (level 2), the Chi-square likelihood–ratio test was used to assess the difference between models. The p-values were estimated using Wald statistics and $p < 0.05$ was taken to indicate statistical significance. The statistical analyses were performed using Stata 14 (Stata Corp, College Station, TX). The "svy" set command was used to adjust for the complex sampling design used by TDHS-MIS. All estimates were weighted to correct for non-responses and disproportionate sampling. To assess potential multicollinearity among the independent variables included in the final regression model, we performed a multicollinearity diagnostic using the Variance Inflation Factor (VIF All variables—both continuous and categorical (entered as dummy variables with appropriate reference categories)—were included in the assessment. A commonly accepted threshold for concern is a VIF value exceeding 5. In our analysis, all variables exhibited VIF values below 3.0, indicating no evidence of significant multicollinearity and supporting the stability and robustness of the regression estimates.

To evaluate the spatial distribution of breast cancer screening uptake across regions in Tanzania, Global Moran's I statistic was applied using Python-based geospatial analysis tools. A spatial weights matrix was generated to define the spatial relationships among regions based on relative proximity. The analysis tested the null hypothesis that screening uptake was randomly distributed in space. The result showed a Moran's I value of −0.0065 with a corresponding z-score close to zero, indicating no statistically significant spatial clustering. This suggests that the observed regional variation in screening uptake was largely random and not driven by spatial proximity.

### Ethical considerations

The 2022 TDHS-MIS was approved by the Tanzania National Institute for Medical Research (NIMR), the Zanzibar Medical Ethics and Research Committee (ZAMREC), and the Institutional Review Board of ICF International in the USA. The informed consent was requested and obtained from the respondents after adequately explaining all relevant aspects of the study, including its aim and interview procedures. All respondents, who accepted to participate in the surveys were provided with a signed written informed consent. In this accord, the current study was based on an analysis of the existing public domain of the 2022 TDHS-MIS dataset which is freely available online and with all participants' names or identifiers information detached. Therefore, the ethical approval for the current analysis was automatically deemed unnecessary. Moreover, permission to use the aforementioned dataset used in this study is obtained from the DHS Program accessed through https://dhsprogram.com/data/new-user-registration.cfm.

### Results

#### Baseline characteristics of the respondents

Table 1 presents the baseline characteristics of the respondents in which a total of 15254 women between 15 and 49 years old were interviewed and included in the analysis. The median age (IQR) of the respondents was 28 (21–37) years. About 61% of the respondents were living with their spouse at the time of the interview. More than 15% did not attend any formal education. More than one-third were not employed in the last 12 months before the interview and only 6% reported having any type of health insurance. Almost 60% of the respondents were multiparas and less than one-tenth reported reading newspapers/magazines at least once a week while nearly 13% reported using the internet in the last 12 months before the interview. About one-third of the respondents were from communities in households with poor wealth status while nearly two-thirds of respondents were from rural residences.

 

**Table 1. Percent distribution of women of reproductive age by selected background characteristics, TDHS (n = 15 254).**

| Variable | n (%) weighted |
|---|---|
| **Level 1 (Individual) variables** | |
| **Age** (median (IQR)=28 (21–37)) | |
| 15-29 | 8343 (54.69) |
| 30-34 | 2076 (13.61) |
| 35-39 | 1884 (12.35) |
| 40-49 | 2951 (19.34) |
| **Marital status** | |
| Never married | 4047 (26.53) |
| Married/living together | 9252 (60.65) |
| Separated/widow/divorced | 1955 (12.82) |
| **Education level** | |
| None | 2450 (16.06) |
| Primary | 8124 (53.25) |
| Secondary | 4467 (29.29) |
| Tertiary | 213 (1.40) |
| **Employed in last 12 months** | |
| Not employed | 5452 (35.74) |
| Employed for cash | 6262 (41.05) |
| Employed but paid-in-kind | 3540 (23.21) |
| **Covered by health insurance** | |
| No | 14366 (94.18) |
| Yes | 888 (5.82) |
| **Parity** (median (IQR)=2 (0–4)) | |
| Nullipara | 3873 (25.39) |
| Primipara | 2291 (15.02) |
| Multipara | 9090 (59.59) |
| **Reading newspaper/magazine** | |
| No | 14286 (93.66) |
| Yes | 968 (6.34) |
| **Listening to radio** | |
| No | 10385 (68.08) |
| Yes | 4869 (31.92) |
| **Watching television** | |
| No | 10707 (70.19) |
| Yes | 4547 (29.81) |
| **Use of internet in last 12 months** | |
| No | 13304 (87.22) |
| Yes | 1950 (12.78) |
| **Level 2 (cluster) variables** | |
| **Household wealth status** | |
| Poor | 5044 (33.06) |
| Middle | 2881 (18.88) |
| Rich | 7329 (48.05) |
| **Geographical zone** | |
| Central | 1156 (7.58) |
| Coastal | 3415 (22.39) |

*(Continued)*

**Table 1.** (Continued)

| Variable | n (%) weighted |
|---|---|
| Lake | 3920 (25/70) |
| Northern | 1393 (9.13) |
| Southern | 1187 (7.78) |
| Southern Highlands | 1668 (10.93) |
| Western | 1801 (11.81) |
| Zanzibar | 714 (4.68) |
| **Residence** | |
| Rural | 9808 (64.30) |
| Urban | 5446 (35.70) |

### Uptake of breast cancer screening

Fig 2 shows the percentage distribution of uptake of breast cancer screening among women of reproductive age in Tanzania. The findings revealed that only 5% of the respondents reported to uptake screening for breast cancer.

### Uptake of breast cancer screening by zones and regions

Fig 3 presents the proportion of uptake of breast cancer screening by zones among women of reproductive health in Tanzania. The results indicated that the uptake of breast cancer screening was less than 10% in all zones of Tanzania. The lowest uptake was in the Western (2.17%) and Southern (3.34) zones of Tanzania. After unpacking the zones into regions, it was observed that the regions with the poorest uptake of breast cancer screening were Kigoma (1.68%), Katavi (1.94%), Singida (1.54%), and Tabora (1.66%) as shown in Fig 4.

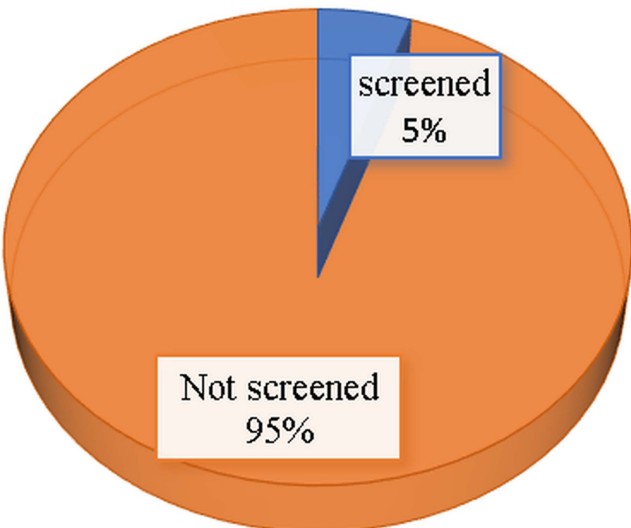

**Fig 2. Percentage of uptake of breast cancer among women of reproductive age, TDHS-MIS 2022 (n = 15 254).**

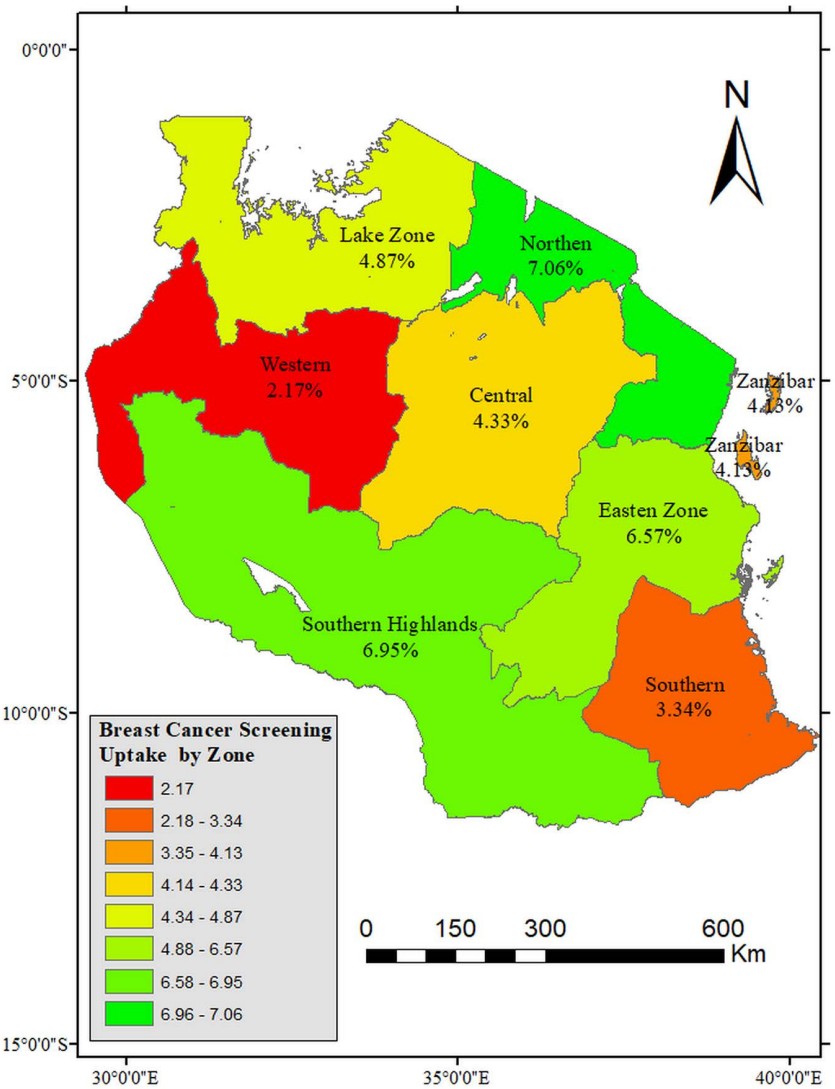

**Fig 3. Proportion distribution of uptake of breast cancer screening among women of reproductive age by zones, TDHS-MIS 2022 (n = 15 254).**
**Shapefiles Source:** https://www.nbs.go.tz/statistics/topic/gis.

## Spatial autocorrelation results

The Global Moran's I statistic was used to assess the spatial autocorrelation of breast cancer screening uptake across Tanzanian regions. The analysis yielded a Moran's I value of −0.0065, with an expected value under spatial randomness of −0.0333, and a corresponding z-score approximately equal to zero. These results indicate that the spatial distribution of screening uptake was not significantly clustered or dispersed, but rather followed a largely random spatial pattern. The absence of statistically significant spatial autocorrelation suggests that neighboring regions did not exhibit similar screening uptake rates. This finding was further supported by visual inspection of the Moran interpretation curve, which positioned the observed value near the center of the distribution, consistent with spatial randomness (Fig 5).

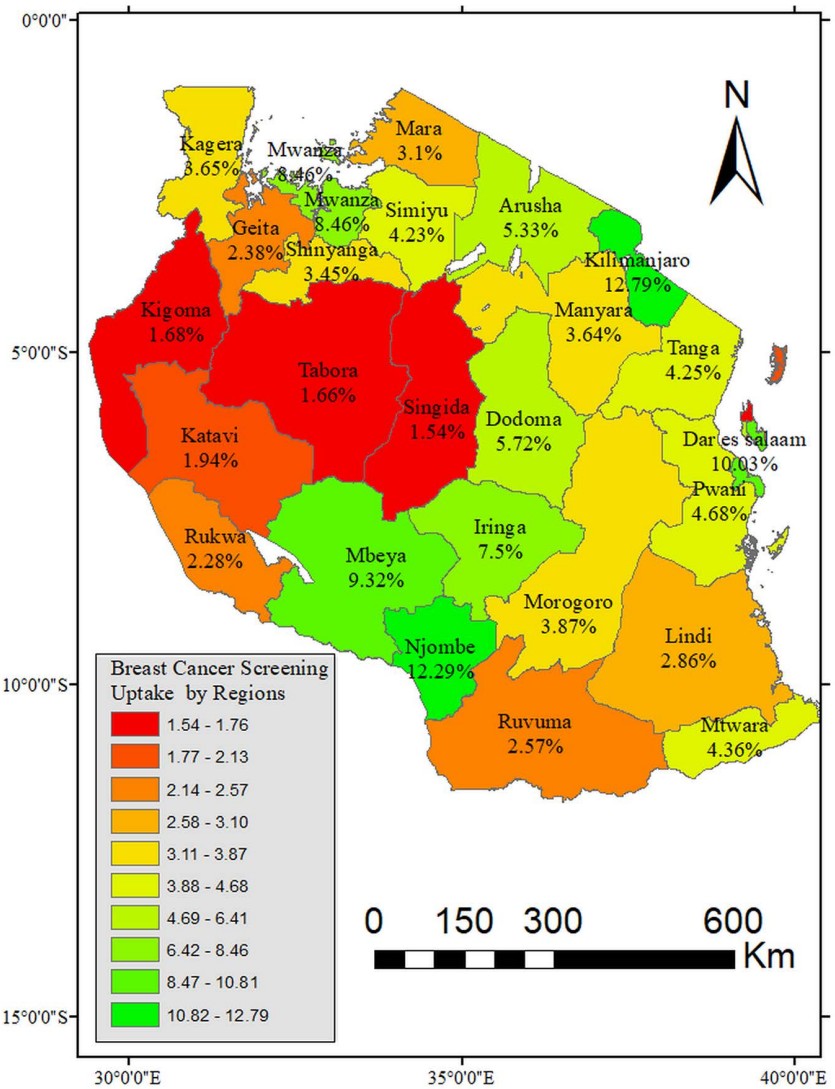

**Fig 4. Proportion distribution of uptake of breast cancer screening among women of reproductive age by regions, TDHS-MIS 2022 (n = 15 254).** Shapefiles Source: https://www.nbs.go.tz/statistics/topic/gis.

## Determinants of low uptake of breast cancer screening

Table 2 presents the results of a mixed-effects multilevel logistic regression of the association between the outcome variable (low uptake of breast cancer screening) and several identified factors as independent variables.

## Fixed effect component results

The final model IV which includes the individual level and community (cluster) level variables shows the factors associated with the uptake of breast cancer screening among women of reproductive age in Tanzania. The findings revealed that the odds of uptake of breast cancer screening were three times higher among older women aged 40–49 years [AOR = 2.88, 95%CI; 2.10–3.95] compared to younger women aged 15–29 years. Women with primary [AOR = 1.80, 95%CI; 1.22–2.65], secondary [AOR = 2.34, 95%CI; 1.51–3.63], and tertiary [AOR = 4.51, 95%CI; 2.33–8.75] education had higher odds

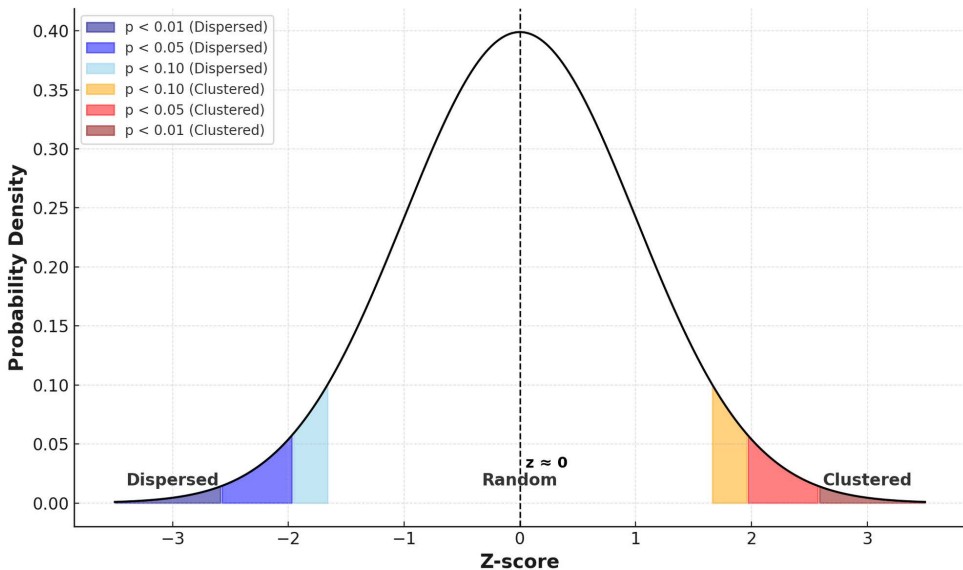

**Fig 5. Spatial distribution of uptake of breast cancer screening among women of reproductive age in Tanzania. Source:** Authors' construct based on the 2022 TDHS-MIS data.

of uptake breast cancer screening compared with those without any formal education. Also, women who were covered with any type of health insurance [AOR = 2.04, 95%CI; 1.54–2.70] and multipara [AOR = 2.21, 95%CI; 1.52–3.21] had higher odds of uptake breast cancer screening compared with their counterparts. In addition, women who read newspapers or magazines [AOR = 1.49, 95%CI; 1.09–2.05] and those who reported using the internet [AOR = 1.36, 95%CI; 1.03–1.79] had higher odds of uptake breast cancer screening compared to their counterparts. Women from the northern zone [AOR = 1.81, 95%CI; 1.06–3.08] and urban residents [AOR = 1.63, 95%CI; 1.28–2.07] had higher odds of uptake of breast cancer screening compared to those from the central zone and rural residence respectively.

### Random effect component results

The random effects model I (empty model) showed significant variability in the odds of uptake of breast cancer screening between clusters [σ2 = 0.75, 95%CI; 0.57–0.98]. Furthermore, the model revealed that nearly 20% of uptake of breast cancer screening was attributed to the variations between the clusters [ICC = 0.19, 95%CI; 0.15–0.23].

### Discussion

This study investigated the spatial distribution and determinants of low uptake of breast cancer screening among women of reproductive age. While other studies looked at these determinants, they focused on only health facilities or regional levels and thus did not account for the spatial differences in the country.

The results support evidence of very low uptake of breast cancer screening in Tanzania. In all zones of Tanzania, the uptake of breast cancer screening was less than 10%. However, it was the lowest in the Western Zone followed by the Southern Zone. Further, the regions with the poorest uptake of breast cancer screening were Kigoma, Katavi, Singida, and Tabora. Throughout Tanzania, there has been a low uptake of breast cancer screening. Most studies that were done in pockets of Africa specifically Tanzania, show a low uptake of breast cancer screening. Screening is an important aspect of reducing mortality in resource-limited areas because treatment can be expensive and too late. Uptake of screening has been shown to reduce mortality and improve survival rate when cancer is detected early.

**Table 2.** Multilevel logit models for the factors associated with breast cancer screening among women of reproductive age, TDHS 2022 (n = 15 254).

| Variable | Model 1 AOR [95% CI] | Model 2 AOR [95% CI] | Model 3 AOR [95% CI] | Model 4 AOR [95% CI] |
|---|---|---|---|---|
| *Fixed Effects Component* | | | | |
| **Level 1 (individual) variables** | | | | |
| **Age** (ref: 15–29) | | | | |
| 30-34 | | 1.33 [0.94-1.88] | | 1.27 [0.90-1.79] |
| 35-39 | | 2.13 [1.47-3.08] | | 1.97 [1.36-2.84] |
| 40-49 | | 3.11 [2.27-4.27] | | 2.88 [2.10-3.95] |
| **Marital status** (ref: Never married) | | | | |
| Married/living together | | 1.25 [0.85-1.83] | | 1.30 [0.89-1.89] |
| Separated/widow/divorced | | 1.52 [0.93-2.47] | | 1.57 [0.98-2.54] |
| **Education level** (ref: None) | | | | |
| Primary | | 2.27 [1.56-3.30] | | 1.80 [1.22-2.65] |
| Secondary | | 3.24 [2.10-5.00] | | 2.34 [1.51-3.63] |
| Tertiary | | 6.19 [3.17-12.08] | | 4.51 [2.33-8.75] |
| **Employed in last 12 months** (ref: Not employed) | | | | |
| Employed for cash | | 1.33 [1.01-1.73] | | 1.26 [0.97-1.64] |
| Employed but paid-in-kind | | 0.91 [0.66-1.27] | | 1.00 [0.72-1.39] |
| **Covered by health insurance** (ref: No) | | | | |
| Yes | | 2.05 [1.54-2.73] | | 2.04 [1.54-2.70] |
| **Parity** (ref: Nullipara) | | | | |
| Primipara | | 1.64 [1.13-2.38] | | 1.60 [1.10-2.33] |
| Multipara | | 2.20 [1.51-3.19] | | 2.21 [1.52-3.21] |
| **Reading newspaper/magazine** (ref: No) | | | | |
| Yes | | 1.52 [1.11-2.08] | | 1.49 [1.09-2.05] |
| **Listening to radio** (ref: No) | | | | |
| Yes | | 1.17 [0.93-1.47] | | 1.11 [0.89-1.40] |
| **Watching television** (ref: No) | | | | |
| Yes | | 1.23 [0.97-1.58] | | 0.99 [0.78-1.26] |
| **Use of internet in last 12 months** (ref: No) | | | | |
| Yes | | 1.53 [1.15-2.03] | | 1.36 [1.03-1.79] |
| **Level 2 (cluster) variables** | | | | |
| **Household wealth status** (ref: poor) | | | | |
| Middle | | | 2.03 [1.45-2.84] | 1.81 [1.29-2.54] |
| Rich | | | 3.05 [2.23-4.18] | 2.20 [1.57-3.09] |
| **Geographical zone** (ref: Central) | | | | |
| Coastal | | | 1.21 [0.74-1.97] | 1.28 [0.80-2.07] |
| Lake | | | 1.13 [0.69-1.85] | 1.21 [0.74-2.00] |
| Northern | | | 1.96 [1.15-3.32] | 1.81 [1.06-3.08] |
| Southern | | | 0.97 [0.56-1.69] | 1.00 [0.56-1.77] |
| Southern Highlands | | | 1.81 [1.10-2.97] | 1.61 [0.98-2.66] |
| Western | | | 0.71 [0.42-1.21] | 0.86 [0.50-1.46] |
| Zanzibar | | | 0.92 [0.55-1.54] | 0.95 [0.57-1.59] |
| **Residence** (ref: Rural) | | | | |
| Urban | | | 1.74 [1.39-2.18] | 1.63 [1.28-2.07] |
| *Random Effects Component* | | | | |

*(Continued)*

**Table 2.** (Continued)

| Variable | Model 1 AOR [95% CI] | Model 2 AOR [95% CI] | Model 3 AOR [95% CI] | Model 4 AOR [95% CI] |
|---|---|---|---|---|
| Level 2 (community) variance | 0.75 [0.57-0.98] | 0.36 [0.24-0.54] | 0.27 [0.17-0.42] | 0.24 [0.15-0.40] |
| Intraclass correlation | 0.19 [0.15-0.23] | 0.10 [0.07-0.14] | 0.08 [0.05-0.11] | 0.07 [0.04-0.11] |
| Akaike's information criterion | 6030.12 | 5453.34 | 5838.56 | 5371.071 |
| Log pseudolikelihood | −3013.06 | −2707.67 | −2907.28 | −2656.54 |

However, this is not the case in Tanzania, and to this end, severe outcomes of breast cancer and high mortality prevail. This is because of low breast cancer screening uptake resulting in late detection of cancer and late presentation at health facilities for care.

Further, the study shows that the lowest uptake of breast cancer screening occurred in the Western and Southern zones of Tanzania, with Kigoma, Katavi, Singida, and Tabora.being the regions with the poorest performance. These regions are predominantly rural and fac multiple systemic challenges that hinder access to preventive health services. Key barriers include limited availability of health facilities, long distances to care centers, and shortage of well-trained healthcare providers who can support such health promotion (breast cancer screening) [31]. In particular, Kigoma and Katavi are among the regions with the lowest health workforce density in the country, which contributes to reduced service delivery capacity. Additionally, cultural norms and lower levels of health awareness in these areas may influence women's health-seeking behavior [32]. Weak referral systems and underdeveloped diagnostic infrastructure further delay early detection and treatment [33]. Similar challenges have been reported in cervical cancer screening programs, where regional disparities in resource allocation and outreach efforts have contributed to suboptimal screening coverage [34]. These findings highlight the broader issue of inequitable distribution of healthcare resources in Tanzania, where emphasis remains largely on curative rather than preventive services, ultimately leaving some regions underserved and at higher risk of poor cancer outcomes [34–36].

While the multilevel analysis revealed substantial between-cluster variability in uptake of breast cancer screening, as indicated by an ICC of 0.19, the spatial autocorrelation test (Global Moran's I = −0.0065) indicated that this variation was not significantly clustered in space. This suggests that, although differences in screening uptake exist across clusters, they are not driven by spatial proximity. Instead, the disparities likely reflect underlying contextual and programmatic differences such as health infrastructure, awareness, access to services, and socioeconomic conditions. These insights emphasize the need for region-specific strategies that go beyond geographic targeting and instead address the contextual barriers affecting screening uptake.

The study reveals that older women, those with formal education, and multiparous women had significantly higher odds of undergoing breast cancer screening. Additionally, health insurance coverage, reading newspapers or magazines, and using the Internet were also associated with increased uptake. Older women are more likely to accumulate health-knowledge over time and can demonstrate greater autonomy and freedom in seeking healthcare services. This aligned with previous findings in sub-Saharan Africa indicating that age is positively associated with awereness nd engagement in preventive health behaviors, including cancer screening [37,38].

Women with formal education were also more likely to participate in breast cancer screening compared to their counterparts without formal education. This finding is consistent with several studies conducted in Tanzania and other low- and middle-income countries (LMICs), which have reported that formal education attainment enhance health literacy, decision-making capacity, and health service utilization including cancer screening [25,39–42]. For example, a study in Ethiopia and another in Kenya similarly found that women with primary, secondary or higher education were significantly more likely to undergo breast cancer screening than uneducated women [43,44].

Furthermore, having any form of health insurance significantly increased the likelihood of cancer screening uptake [45]. Insurance facilitates financial access to health services and reduces the out-of-pocket burden, which is a critical barrier in resource-limited settings. This is corroborated by studies from other LMICs, including Burkina Faso, Ivory Coast, Kenya, Namibia and Nigeria, where women with health insurance were found to be more likely to utilize preventive services, including mammography and clinical breast examinations [13,46,47].

Lastly, access to information emerged as a key determinant of breast cancer screening uptake consistent with findings from prior research [48]. Our study has shown that women who are reading newspapers or magazines or are used the internet had significantly higher odds takeoff undergoing breast cancer screening. This finding echoes previous research showing that media exposure plays a vital role in disseminating health information, shaping perceptions, and motivating health-seeking behavior [6,11,49]. Moreover, broader evidence from LMICs, including Bangladesh, highlights that regular exposure to media significantly enhances the use of reproductive and preventive health services [50]. These findings suggest that strengthening media-based health communication strategies could be instrumental in increasing screening uptake, particularly in underserved populations.

The strength of this study is that, it is the first in the region using recent nationally representative sample to perform geospatial and multilevel analysis. This approach allowing for generalizable conclusions regarding uptake of breast cancer screening in that population of interest. The provided estimates were adjusted and weighted to correct for non- response and disproportionate sampling. However, the study had some limitation such as the use of data from cross-sectional survey meant that causality assumptions could not be inferred. Consequently, the results should be interpreted with caution. Also, as a secondary analysis it faced the problem of missing important variables such as beliefs, myths, attitudes, and culture which might have correlation with uptake of breast cancer screening, therefore, a study that might include these variables is thus recommended.

## Conclusions

In conclusion, this study highlights the persistently low uptake of breast cancer screening in Tanzania, with rural areas exhibiting notably lower rates compared to urban counterparts. Key individual-level barriers include lack of formal education, absence of health insurance coverage, and limited exposure to reliable health information contribute significantly to these disparities. To reduce the observed inequalities across different regions and zones, policymakers should prioritize the equitable allocation of screening resources. This includes implementing community-based educational programs tailored to women with low literacy levels, particularly in rural and underserved areas. Additionally, the deployment of mobile breast cancer screening units could significantly improve access in remote regions where healthcare infrastructure is limited. Furthermore, government and non-governmental organizations should invest in mass media campaigns using newspapers, radio, and digital platforms to raise awareness and promote regular screening, especially in low-uptake regions. Furthermore, future longitudinal studies are recommended to assess the long-term effectiveness and sustainability of these interventions and to guide evidence-based policy adjustments over time.

## Supporting information

**S1 File. Dataset used in this analysis.**
(CSV)

## Acknowledgments

We would like to acknowledge ICF International, Rockville, Maryland, USA, through the DHS program for giving us permission and access to the 2022 TDHS-MIS dataset.

## Author contributions

**Conceptualization:** Deogratius Bintabara, Costantine C. Kamata, Ramadhani Mohamedi, Namanya Basinda.

**Data curation:** Deogratius Bintabara, Ramadhani Mohamedi, Namanya Basinda.

**Formal analysis:** Deogratius Bintabara, Ramadhani Mohamedi, Namanya Basinda.

**Methodology:** Deogratius Bintabara, Costantine C. Kamata, Ramadhani Mohamedi, Namanya Basinda.

**Project administration:** Deogratius Bintabara, Costantine C. Kamata.

**Resources:** Deogratius Bintabara.

**Software:** Deogratius Bintabara.

**Supervision:** Deogratius Bintabara, Namanya Basinda.

**Validation:** Deogratius Bintabara.

**Visualization:** Deogratius Bintabara.

**Writing – original draft:** Deogratius Bintabara, Costantine C. Kamata, Ramadhani Mohamedi, Namanya Basinda.

**Writing – review & editing:** Deogratius Bintabara, Costantine C. Kamata, Ramadhani Mohamedi, Namanya Basinda.

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
