## [Decision Letter · Decision Letter 0]

11 Mar 2025

Dear Dr. Bintabara,

Thank you for submitting your manuscript to PLOS ONE. After careful consideration, we feel that it has merit but does not fully meet PLOS ONE’s publication criteria as it currently stands. Therefore, we invite you to submit a revised version of the manuscript that addresses the points raised during the review process.

We look forward to receiving your revised manuscript.

Kind regards,

Nontuthuzelo Iris Muriel Somdyala, Ph.D

Academic Editor

PLOS ONE

Journal Requirements:

2. We are unable to open your Supporting Information file Supplementary File 1.rar. Please kindly revise as necessary and re-upload.

3. We note that Figures 3 and 4 to in your submission contain map images which may be copyrighted. All PLOS content is published under the Creative Commons Attribution License (CC BY 4.0), which means that the manuscript, images, and Supporting Information files will be freely available online, and any third party is permitted to access, download, copy, distribute, and use these materials in any way, even commercially, with proper attribution. For these reasons, we cannot publish previously copyrighted maps or satellite images created using proprietary data, such as Google software (Google Maps, Street View, and Earth). For more information, see our copyright guidelines: http://journals.plos.org/plosone/s/licenses-and-copyright .

     1. You may seek permission from the original copyright holder of Figures 3 and 4 to publish the content specifically under the CC BY 4.0 license. 

4. Please include captions for your Supporting Information files at the end of your manuscript, and update any in-text citations to match accordingly. Please see our Supporting Information guidelines for more information: http://journals.plos.org/plosone/s/supporting-information .

Reviewers' comments:

Reviewer's Responses to Questions

**Comments to the Author**

1. Is the manuscript technically sound, and do the data support the conclusions?

Reviewer #1: Yes

Reviewer #2: Yes

2. Has the statistical analysis been performed appropriately and rigorously?

Reviewer #1: Yes

Reviewer #2: Yes

3. Have the authors made all data underlying the findings in their manuscript fully available?

Reviewer #1: No

Reviewer #2: Yes

4. Is the manuscript presented in an intelligible fashion and written in standard English?

Reviewer #1: Yes

Reviewer #2: Yes

Reviewer #1: After reviewing the manuscript titled "Spatial distribution and determinants of low uptake of breast cancer screening among women of reproductive age: a mixed-effect multilevel analysis of Tanzanian population-based survey," here are some constructive comments and suggestions:

Strengths

Important Topic: The study addresses a critical public health issue by exploring the low uptake of breast cancer screening in Tanzania. This focus is highly relevant, given the high mortality rates associated with late detection in low-income settings.

Comprehensive Analysis: The use of multilevel logistic regression and spatial analysis offers valuable insights into both individual and contextual determinants of breast cancer screening uptake.

Policy Implications: The results are practical for policymakers, providing actionable insights to improve screening access and educational outreach, particularly for rural and low-resource areas.

Areas for Improvement

1. Clarity and Flow of Introduction

Suggestion: Clarify the need for spatial analysis in this context and the importance of addressing determinants at both the individual and community levels. This would enhance the introduction's cohesion and clarify the study’s novelty.

Rationale: Although the study motivation is sound, a more specific linkage between the identified problem and the methods used would improve reader engagement.

2. Methodological Details

Sampling Details: More information on how clusters and individuals within clusters were selected would provide a clearer picture of the study’s representativeness.

Confounders and Multicollinearity: The paper briefly mentions VIF checks for multicollinearity. Expanding on this with details about specific variables checked and results would strengthen the readers' confidence in the robustness of findings.

Geospatial Analysis: It’s unclear whether spatial autocorrelation was tested or controlled for in the geospatial analysis. Discussing this aspect would add depth to the spatial findings.

3. Interpretation of Results

Regional Variations: The results mention regional disparities but could better discuss why specific regions (e.g., Kigoma, Katavi) show particularly low screening uptake.

Education and Insurance Findings: These determinants are mentioned, but a comparison with other studies, especially those conducted in similar contexts, would deepen the discussion and contextualize the findings.

Policy Implications: Suggestions for improving screening uptake should be more specific and actionable. For example, detailing how educational programs or mobile screening services could be prioritized for low-uptake regions would add practical value.

4. Figures and Tables

Figures: Some maps and figures could benefit from clearer labeling to make spatial differences in screening uptake more evident at a glance.

Tables: Reorganizing tables, such as separating individual-level and cluster-level predictors, may improve readability.

5. Conclusion and Recommendations

Strengthen Recommendations: The conclusion could include more specific, actionable recommendations, such as strategies for integrating breast cancer screening into existing health outreach programs or leveraging mobile technology.

Long-Term Research Needs: Adding a brief mention of the importance of longitudinal studies to monitor the effectiveness of interventions would provide a forward-looking perspective.

6. Minor Stylistic and Language Edits

The language could be more concise in places, which would improve readability and accessibility for an international audience.

Overall Assessment

The manuscript presents a strong foundation with important findings that have practical implications. Addressing the points above, particularly regarding clarity, methodological transparency, and actionable recommendations, would significantly enhance the paper's impact. Finally ,please make sure that all tables and diagrams are included.

Reviewer #2: The article is very well articulated, However the following could further strengthen the manuscript

Critique

Abstract:

1. Methods: The author should consider including sample size, study population what statistical package did they use, what measure of associated used and confidence level.

2. Results: The author should consider omitting sample size, study population in results section in Abstract

Introduction

3. The study could be more informative if the author could include the Tanzanian government policy towards breast cancer screening but also what has been done so far in the country

4..While describing the Literature review the author grouped factors into socio-cultural, religious, and economic factors, However it could be more informative if all specific factors could be mentioned.

5. The author mentions that there are only health facility level or regional level studies done, and thus calling for a larger-scale

study in the country BUT there is a national-wise study done in Tanzania using same 2022 data (TDHS), the author should do a thorough literature search

Methods

6.study design: The author should clearly state that it was analytical cross-sectional

7.Dependent variable: The author should state how was the coding done ( How was yes and no coded in the STATA)

8.Independent variables: On what grounds did the author categorize variables such as age

On every variable author should provide clear citation what is their reference for categorizing such a variable, Otherwise variables such as age and employment should not have been categorized that way

9. Analysis- The author should clearly state why was Median and IQR used, was there any test used ?

10. What does cluster level 2 mean to author ? while level 1 is clearly mentioned (individuals or households) level 2 has remained silent

Results and discussion: very well written

**Do you want your identity to be public for this peer review?** For information about this choice, including consent withdrawal, please see our Privacy Policy

Reviewer #1: **Yes: ** Abraham Keffale

Reviewer #2: No

---

## [Author Response · Author response to Decision Letter 1]

2 Jun 2025

Response to Editorial requests and reviewers’ comments

Editorial request

Response: Thank you for the reminder. We have reviewed the PLOS ONE style guidelines and have revised the manuscript and all associated files to ensure full compliance, including formatting and file naming conventions.

2. We are unable to open your Supporting Information file Supplementary File 1.rar. Please kindly revise as necessary and re-upload.

Response: Thank you for notifying us. We apologize for the inconvenience caused by the inaccessible file. We have now extracted and converted the contents of Supplementary File 1.rar into standard, accessible formats

3. We note that Figures 3 and 4 to in your submission contain map images which may be copyrighted. All PLOS content is published under the Creative Commons Attribution License (CC BY 4.0), which means that the manuscript, images, and Supporting Information files will be freely available online, and any third party is permitted to access, download, copy, distribute, and use these materials in any way, even commercially, with proper attribution. For these reasons, we cannot publish previously copyrighted maps or satellite images created using proprietary data, such as Google software (Google Maps, Street View, and Earth). For more information, see our copyright guidelines: http://journals.plos.org/plosone/s/licenses-and-copyright.

Response: Thank you for your guidance. We confirm that the figures included in the manuscript are original visualizations created by the authors using publicly available microdata from the 2022 Tanzania Demographic and Health Survey and Malaria Indicator Survey (TDHS-MIS). These figures do not reproduce or adapt any copyrighted materials, logos, or pre-published visuals. We have accordingly added the following attribution to each relevant figure:

Source: Authors’ construct based on the 2022 Tanzania Demographic and Health Survey and Malaria Indicator Survey (TDHS-MIS) data.

We trust this satisfies the licensing requirements under the CC BY 4.0 license. Please let us know if any additional clarification is needed.

Reviewer #1:

After reviewing the manuscript titled "Spatial distribution and determinants of low uptake of breast cancer screening among women of reproductive age: a mixed-effect multilevel analysis of Tanzanian population-based survey," here are some constructive comments and suggestions:

Strengths

Important Topic: The study addresses a critical public health issue by exploring the low uptake of breast cancer screening in Tanzania. This focus is highly relevant, given the high mortality rates associated with late detection in low-income settings.

Comprehensive Analysis: The use of multilevel logistic regression and spatial analysis offers valuable insights into both individual and contextual determinants of breast cancer screening uptake.

Policy Implications: The results are practical for policymakers, providing actionable insights to improve screening access and educational outreach, particularly for rural and low-resource areas.

Response: We thank the reviewer for their positive feedback. We appreciate the recognition of our study's relevance to public health, the appropriateness of our multilevel and spatial analysis methods, and the potential policy implications of our findings. These affirm our objective to inform targeted interventions that improve breast cancer screening uptake in underserved areas of Tanzania.

Areas for Improvement

1. Clarity and Flow of Introduction

Suggestion: Clarify the need for spatial analysis in this context and the importance of addressing determinants at both the individual and community levels. This would enhance the introduction's cohesion and clarify the study’s novelty.

Response: Thank you, we revised the introduction to highlight the relevance of spatial and multilevel analyses in identifying geographic and contextual disparities, thereby strengthening the study’s rationale and novelty [Page 6, Line 10-22 to Page 7, Line 5-9].

Rationale: Although the study motivation is sound, a more specific linkage between the identified problem and the methods used would improve reader engagement.

Response: Thank you for the comment, we revised the rationale to clearly link the public health problem of low breast cancer screening uptake with our use of spatial and multilevel analysis [Page 7, Line 5-9].

2. Methodological Details

Sampling Details: More information on how clusters and individuals within clusters were selected would provide a clearer picture of the study’s representativeness.

Response: Thank you for this valuable comment. Details on the two-stage cluster sampling process were added, including how clusters were selected using probability proportional to size and how households were systematically sampled, to clarify the representativeness of the study [see Page 8, Line, Line 19-21 to Page 9, Line 1-18].

Confounders and Multicollinearity: The paper briefly mentions VIF checks for multicollinearity. Expanding on this with details about specific variables checked and results would strengthen the readers' confidence in the robustness of findings.

Response: Thank you for the helpful comment. We have revised the Methods section to provide more detail on the multicollinearity assessment. Specifically, we now clarify that all independent variables included in the final regression model were evaluated using the Variance Inflation Factor (VIF) [see Page 13, Line 1-8].

Geospatial Analysis: It’s unclear whether spatial autocorrelation was tested or controlled for in the geospatial analysis. Discussing this aspect would add depth to the spatial findings.

Response: Thank you for the comments. We clarified that Global Moran’s I was used to assess spatial autocorrelation, which showed no significant clustering. This finding, now discussed in the manuscript, suggests that regional differences in screening uptake are due to contextual factors rather than geographic proximity. [see Page 13, Line 9-15; Page 18, Line 5-14 and Page 24, Line 21-22 to Page 25, Line 1-5].

3. Interpretation of Results

Regional Variations: The results mention regional disparities but could better discuss why specific regions (e.g., Kigoma, Katavi) show particularly low screening uptake.

Response: Thank you for this insightful comment. We have revised the Discussion section to provide a more detailed explanation of the regional disparities in breast cancer screening uptake. Specifically, we now elaborate on why regions such as Kigoma, Katavi, Singida, and Tabora exhibit particularly low screening rates [Page 23, Line 18-22 to Page 24, Line 1-18].

Education and Insurance Findings: These determinants are mentioned, but a comparison with other studies, especially those conducted in similar contexts, would deepen the discussion and contextualize the findings.

Response: Thank you for the comments. We have expanded the discussion on education and insurance by comparing our findings with studies from similar low- and middle-income settings. This addition provides a more contextualized and comprehensive interpretation of our results [Page 25, Line 17-23 to Page 26, Line 1-11].

Policy Implications: Suggestions for improving screening uptake should be more specific and actionable. For example, detailing how educational programs or mobile screening services could be prioritized for low-uptake regions would add practical value.

Response: Thank you for your insightful suggestion. In response, we have revised the conclusion to include more specific and actionable policy recommendations [Page 27, Line 14-22 to Page 28, Line 1-8].

4. Figures and Tables

Figures: Some maps and figures could benefit from clearer labeling to make spatial differences in screening uptake more evident at a glance.

Response: Thank you for this valuable feedback. In response, we have improved the clarity of the maps and figures by enhancing the labeling of geographical areas and adjusting the legends to make spatial differences in screening uptake more visually evident [see Fig 3 and 4].

Tables: Reorganizing tables, such as separating individual-level and cluster-level predictors, may improve readability.

Response: Thank you for your helpful suggestion. In response, we have reorganized the presentation of results by separating individual-level and cluster-level predictors as suggested [see Table 1 and 2].

5. Conclusion and Recommendations

Strengthen Recommendations: The conclusion could include more specific, actionable recommendations, such as strategies for integrating breast cancer screening into existing health outreach programs or leveraging mobile technology.

Response: Thank you for your insightful suggestion. In response, we have revised the conclusion to include more specific and actionable policy recommendations [Page 27, Line 14-23 to Page 28, Line 1-8].

Long-Term Research Needs: Adding a brief mention of the importance of longitudinal studies to monitor the effectiveness of interventions would provide a forward-looking perspective.

Response: Thank you for your thoughtful suggestion. In response, we have revised the conclusion to include a forward-looking statement emphasizing the importance of future longitudinal studies. [Page 28, Line 8-10]

6. Minor Stylistic and Language Edits

The language could be more concise in places, which would improve readability and accessibility for an international audience.

Response: Thank you for this important observation. We have carefully reviewed the manuscript and revised several sections to improve clarity, eliminate redundancy, and enhance overall conciseness.

Overall Assessment

The manuscript presents a strong foundation with important findings that have practical implications. Addressing the points above, particularly regarding clarity, methodological transparency, and actionable recommendations, would significantly enhance the paper's impact. Finally ,please make sure that all tables and diagrams are included.

Response: We sincerely thank the reviewer for the positive evaluation of our manuscript and for acknowledging the importance of our findings. We have carefully addressed all the specific comments provided, with particular attention to improving clarity, methodological transparency, and the inclusion of more specific and actionable recommendations. Additionally, we have ensured that all tables and figures are complete, clearly labeled, and appropriately referenced within the text. We believe these revisions have significantly strengthened the manuscript and enhanced its relevance and impact.

Reviewer #2:

General comment: The article is very well articulated, However the following could further strengthen the manuscript

Response: We sincerely appreciate the reviewer’s positive feedback on the quality and clarity of the manuscript. We also thank you for the constructive comments provided. Each suggestion has been carefully considered and addressed to enhance the rigor and presentation of the study.

Critique

Abstract:

Comment #1. Methods: The author should consider including sample size, study population what statistical package did they use, what measure of associated used and confidence level.

Response: Thank you for this valuable suggestion. We have revised the Methods section of the Abstract to include the total sample size, the study population, the statistical software used, and the measures of association (adjusted odds ratios with 95% confidence intervals) [Page 2, Line 10-19].

Comment #2. Results: The author should consider omitting sample size, study population in results section in Abstract

Response: Thank you for the suggestion. In response, we have revised the Results section of the Abstract by omitting the sample size and study population details, as these are already stated in the Methods section.

Introduction

Comment #3. The study could be more informative if the author could include the Tanzanian government policy towards breast cancer screening but also what has been done so far in the country

Response: Thank you for this valuable suggestion. In response, we have revised the manuscript to include a description of the Tanzanian government's policy measures and initiatives aimed at promoting breast cancer screening [Page 5, Line 5-13].

Comment #4. While describing the Literature review the author grouped factors into socio-cultural, religious, and economic factors, However it could be more informative if all specific factors could be mentioned.

Response: Thank you for this helpful comment. In response, we have revised the relevant paragraph in the Introduction to include more specific examples of the socio-cultural, religious, and economic factors influencing the low uptake of breast cancer screening [Page 5, Line 14-23 to Page 6, Line 1-4].

Comment #5. The author mentions that there are only health facility level or regional level studies done, and thus calling for a larger-scale study in the country BUT there is a national-wise study done in Tanzania using same 2022 data (TDHS), the author should do a thorough literature search.

Response: Thank you for this important observation. In response to your comment, we have revised the manuscript to clarify that while national data have been utilized in previous studies, our study adds further value by applying both spatial analysis and mixed-effect multilevel modeling to uncover geographic disparities and hierarchical determinants of screening uptake [Page 6, Line 8-12].

Methods

Comment #6. Study design: The author should clearly state that it was analytical cross-sectional

Response: Thank you for this helpful comment. We have revised the manuscript to explicitly state that the study employed an analytical cross-sectional design [Page 8, Line 13-14].

Comment #7. Dependent variable: The author should state how was the coding done ( How was yes and no coded in the STATA)

Response: Thank you for this important observation. We have revised the Measurement of Variable subsection in the Methods section to clearly describe how the dependent variable was coded in Stata [Page 10, Line 15-20].

Comment #8. Independent variables: On what grounds did the author categorize variables such as age. On every variable author should provide clear citation what is their reference for categorizing such a variable, otherwise variables such as age and employment should not have been categorized that way

Response: Thank you for your insightful comment. In response, we have revised the manuscript to include references that support the categorization of independent variables such as age and employment status [Page 11, Line 12-13].

Comment #9. Analysis- The author should clearly state why was Median and IQR used, was there any test used?

Response: Thank you for your comment. We have revised the Statistical Analysis subsection to clarify that median and interquartile range (IQR) were used to summarize continuous variables (such as age and parity) due to their non-normal distribution [Page 11, Line 18-21].

Comment #10. What does cluster level 2 mean to author? while level 1 is clearly mentioned (individuals or households) level 2 has remained silent

Response: Thank you for pointing this out. We have revised the Statistical Analysis subsection to explicitly define Level 2 in the context of our multilevel analysis [Page 12, Line 2-3].

Comment 11. Results and discussion: very well written

Response: We sincerely appreciate the positive feedback. We are pleased to know that the reviewer found the Results and Discussion sections well written. Thank you for your encouraging remarks.

---

## [Decision Letter · Decision Letter 1]

29 Jul 2025

Dear Dr. Bintabara,

Thank you for submitting your manuscript to PLOS ONE. After careful consideration, we feel that it has merit but does not fully meet PLOS ONE’s publication criteria as it currently stands. Therefore, we invite you to submit a revised version of the manuscript that addresses the points raised during the review process.

We look forward to receiving your revised manuscript.

Kind regards,

Daniele Ugo Tari, M.D.

Academic Editor

PLOS ONE

Journal Requirements:

Reviewers' comments:

Reviewer's Responses to Questions

**Comments to the Author**

Reviewer #2: (No Response)

2. Is the manuscript technically sound, and do the data support the conclusions?

Reviewer #2: Yes

3. Has the statistical analysis been performed appropriately and rigorously?

Reviewer #2: Yes

4. Have the authors made all data underlying the findings in their manuscript fully available?

Reviewer #2: Yes

5. Is the manuscript presented in an intelligible fashion and written in standard English?

Reviewer #2: Yes

Reviewer #2: Dear authors,

Thank you for re-submitting this very important and improved manuscript

Most of the comments have been adequately addressed however the following comments were not properly addressed

1. Regarding the previously published you were requested to do thorough literature search as there there are studies done in Tanzania using the same TDHS data and covered the same topic on breast cancer screening, although spatial distribution and multilevel is your go to gap but there is a need of acknowledging those studies and state that you are bringing spatial distribution and multilevel on the table

2. Is there any justification that you categorized Employment the way you did? I would advice (Employed Vs Unemployed)

Otherwise the manuscript looks good to me.

Wishing you all the best

**Do you want your identity to be public for this peer review?** For information about this choice, including consent withdrawal, please see our Privacy Policy

Reviewer #2: No

---

## [Author Response · Author response to Decision Letter 2]

13 Oct 2025

Response to reviewers

Reviewer #2:

General comment: Thank you for re-submitting this very important and improved manuscript. Most of the comments have been adequately addressed however the following comments were not properly addressed

Response: Thank you for the careful review and for acknowledging the improvements to our manuscript. We appreciate the additional guidance. We have now revisited the points identified as not fully addressed and made targeted revisions accordingly.

Comment 1. Regarding the previously published you were requested to do thorough literature search as there there are studies done in Tanzania using the same TDHS data and covered the same topic on breast cancer screening, although spatial distribution and multilevel is your go to gap but there is a need of acknowledging those studies and state that you are bringing spatial distribution and multilevel on the table

Response: Thank you for the suggestion. Following an expanded literature search, we evaluated relevant TDHS-based studies on breast cancer screening in Tanzania and have now cited two pertinent references in the manuscript. In line with the editor’s guidance, we cited only works directly relevant to our aims. We also clarified our added value by explicitly stating that prior analyses focused on individual-level determinants, whereas our study integrates spatial distribution and multilevel modeling to capture geographic heterogeneity and community-level effects [See Page 6, Line 1 – 4; and references 26 and 27].

Comment 2. Is there any justification that you categorized Employment the way you did? I would advice (Employed Vs Unemployed)

Response: We appreciate the suggestion to collapse Employment into a binary indicator. However,we retain the DHS employment categories (not employed; employed for cash; employed but paid-in-kind) because they capture meaningful differences in ability-to-pay and access barriers that a simple employed/unemployed split would obscure, and they improve confounding control and policy relevance. This specification also aligns with standard DHS recode variables (v714, v716) for comparability with TDHS literature.

---

## [Decision Letter · Decision Letter 2]

20 Nov 2025

Spatial autocorrelation and determinants of low uptake of breast cancer screening among women of reproductive age: a mixed-effect multilevel analysis of Tanzanian population-based survey

PONE-D-24-43112R2

Dear Dr. Bintabara,

We’re pleased to inform you that your manuscript has been judged scientifically suitable for publication and will be formally accepted for publication once it meets all outstanding technical requirements.

Kind regards,

Daniele Ugo Tari, M.D.

Academic Editor

PLOS ONE

Additional Editor Comments (optional):

Dear Authors,

all the comments have been addressed.

Consequently, I think that the paper can be accepted in present form.

Sincerely,

Reviewers' comments:

Reviewer's Responses to Questions

**Comments to the Author**

Reviewer #2: All comments have been addressed

2. Is the manuscript technically sound, and do the data support the conclusions?

Reviewer #2: Yes

3. Has the statistical analysis been performed appropriately and rigorously?

Reviewer #2: Yes

4. Have the authors made all data underlying the findings in their manuscript fully available?

Reviewer #2: Yes

5. Is the manuscript presented in an intelligible fashion and written in standard English?

Reviewer #2: Yes

Reviewer #2: Dear Authors,

Thank you for addressing all comments. The manuscript looks good. I recommend publication and I wish you all the best

**Do you want your identity to be public for this peer review?** For information about this choice, including consent withdrawal, please see our Privacy Policy

Reviewer #2: No

---

## [Editor Report · Acceptance letter]

PONE-D-24-43112R2

PLOS One

Dear Dr. Bintabara,

I'm pleased to inform you that your manuscript has been deemed suitable for publication in PLOS One. Congratulations! Your manuscript is now being handed over to our production team.

Kind regards,

on behalf of

Dr. Daniele Ugo Tari

Academic Editor

PLOS One